# Characterization of Copper/Zinc Superoxide Dismutase Activity on *Phascolosoma esculenta* (Sipuncula: Phascolosomatidea) and Its Protection from Oxidative Stress Induced by Cadmium

**DOI:** 10.3390/ijms232012136

**Published:** 2022-10-12

**Authors:** Yang Liu, Chen Du, Chenwen Lin, Xinming Gao, Junquan Zhu, Chundan Zhang

**Affiliations:** Key Laboratory of Applied Marine Biotechnology by the Ministry of Education, School of Marine Sciences, Ningbo University, Ningbo 315211, China

**Keywords:** *Phascolosoma esculenta*, bioindicator, detoxification, Cu/Zn SOD

## Abstract

*Phascolosoma esculenta*, an economically important species inhabiting the high tide areas of the intertidal zone, is particularly sensitive to water pollution. Considering its potential as a bioindicator, studies on the ecotoxicology of *P. esculenta* are imperative. The toxic effects of cadmium (Cd) were analyzed by exposing *P. esculenta* to different concentrations of Cd (6, 24, 96 mg/L). In this study, the changes in the antioxidative indexes of total superoxide dismutase (T-SOD), glutathione s-transferase (GST), reduced glutathione (GSH), and microscale malondialdehyde (MDA) were recorded. Copper/zinc superoxide dismutase (Cu/Zn SOD) is one of the most important free radical scavenging members. To reveal the antioxidative function of *P. esculenta,* an important member of the antioxidative system, designated *Pe-Cu/Zn SOD*, was cloned and analyzed. Phylogenic analysis revealed that *Pe-Cu/Zn SOD* was located in the invertebrate evolutionary branch of intracellular Cu/Zn SOD (icCu/Zn SOD). The quantitative real-time polymerase chain reaction results showed that *Pe-Cu/Zn SOD* messenger ribonucleic acid was widely expressed in all tissues examined. The highest expression levels in coelomic fluid after Cd exposure indicated its function in the stress response. Using a prokaryotic expression system, we obtained a *Pe-Cu/Zn SOD* recombinant protein, which enhanced the heavy metal tolerance of *Escherichia coli*. In vivo assays also confirmed that the *Pe**-Cu/Zn SOD* recombinant protein had an antioxidative and free radical scavenging ability. A Cd toxicity experiment, in which purified *Pe-**Cu/Zn SOD* protein was injected into the body cavities of *P. esculenta*, showed that the reactive oxygen species content in the coelomic fluid of the experimental group was significantly lower compared with the control group. These results suggest that *Pe**-Cu/Zn SOD* played a role in Cd detoxification by chelating heavy metal ions and scavenging reactive oxygen free radicals, and that *P. esculenta* could be used as a bioindicator to evaluate heavy metal pollution.

## 1. Introduction

Cadmium (Cd) is a nonessential element that mainly exists in the form of Cd^2+^ in organisms. Its biological toxicity is mainly because of its interference with the various metabolic processes of cells, especially energy metabolism, membrane transport, and protein synthesis. Cd may interfere with the genetic control and repair mechanisms of deoxyribose nucleic acid (DNA) either directly or indirectly. Recently, the mechanisms by which Cd induces oxidative stress have been investigated.

Cadmium has been shown to change the permeability of the mitochondrial membrane, inhibit mitochondrial ATP synthesis, and disturb the mitochondrial electron respiratory transport chain. During this period, Cd induces the accumulation of reactive oxygen species (ROS) [1,2], which refers to the oxygen-containing derivatives of oxygen radicals, including superoxide anions (O_2_^−^) and hydroxyl radicals (OH) [3]. Under normal circumstances, the production and elimination of ROS in organisms is in dynamic equilibrium [4], and moderate amounts of ROS are used as signal transduction molecules [5]. When organisms face heavy metal stress, an overdose of ROS accumulation destroys regulation of the antioxidative system, ultimately causing a series of oxidative damage. In other aspects, Cd can damage the organisms’ antioxidative defense system; for example, Cd could bind with sulfhydryl groups in oxidative enzymes and antioxidative molecules, thereby reducing its ability to scavenge ROS. Cd^2+^ can also replace Zn^2+^, Fe^2+^, and other metal ions in proteins, resulting in an increase in intracellular free metals, which produce excessive ROS through Fenton and Haber–Weiss reactions, and eventually cause lipid peroxidation.

When organisms are in a state of oxidative stress, the antioxidative system is activated to remove excessive ROS [6,7]. Superoxide dismutases (SODs) are important antioxidative enzymes that are widely present in eukaryotes and prokaryotes; they can scavenge excess oxygen free radicals and protect the organism from oxidative damage [8]. According to the different metal auxiliary groups, Cu, Zn, Mn, Fe, and Ni, SODs can be divided into four types: Cu/Zn SOD, Mn-SOD, Fe-SOD, and Ni-SOD [9]. Among these types, Cu/Zn SOD is one of the most important free radical scavenging members and exists primarily in the cytoplasm and intercellular matrix. The Cu/Zn SOD family can be divided into intracellular Cu/Zn SOD (icCu/Zn SOD), encoded by the *SOD1* gene, and extracellular Cu/Zn SOD (ecCu/Zn SOD), encoded by *SOD3* [10]. icCu/Zn SOD is most widely distributed in organisms, mainly in the eukaryotic cytoplasm and also in the chloroplast matrix, peroxisome, and mitochondria [11] while ecCu/Zn SOD is found in the extracellular and cytoplasmic matrices [12]. When organisms are exposed to heavy metals, the expression of Cu/Zn SOD is upregulated to maintain normal physiological functions and scavenge free radicals. For example, after exposing clams to Cd, Fang et al. [13] reported that the expression of Cu/Zn SOD messenger ribonucleic acid (mRNA) was significantly increased in *Mactra veneriformis*, indicating that Cu/Zn SOD protected clams from Cd toxicity. When *Euplotes crassus* was exposed to Cd, antioxidation due to Cu/Zn SOD was observed [14].

*Phascolosoma esculenta* (Sipuncula: Phascolosomatidea) is a Sipuncula that lives in an intertidal flat, and its habitat is easily affected by heavy metal pollution of coastal regions. *P**. esculenta* are sensitive to changes in the environmental conditions, which could reflect the pollution status. In addition, the relative stable life cycle of *P. esculenta* makes this species countable. Thus, *P. esculenta* are a potential bioindicator for monitoring heavy metal pollution in marine mudflats. In order to understand the toxic effect of Cd on *P. esculenta*’s response under Cd stress (*Pe-Cu/Zn SOD*), the acute toxicity of Cd on *P. esculenta* was determined. The effects of Cd on oxidative stress in *P. esculenta* were analyzed by measuring the content or activity of malondialdehyde (MDA) and several important antioxidants, such as SOD, CAT, and reduced glutathione (GSH), in coelomic fluid. Gene cloning, quantitative polymerase chain reaction (qPCR), enzyme activity determination, prokaryotic expression, and flow cytometry were used to analyze the expression and antioxidant function of *Pe-Cu/Zn SOD* under Cd stress. This study provides basic data for the molecular toxicology research of *P.*
*esculenta* and lays a foundation for further studies on *Pe-Cu/Zn SOD* function.

## 2. Results

### 2.1. Response of Antioxidative Indexes in the Supernatants after Cd Exposure

There were no significant differences in the MDA content (Figure 1A) between each time point in the experimental and control groups (*p* > 0.05). In the 6 mg/L group, MDA was significantly increased 96 h after exposure, but no significant differences were noted at any other points in time. When compared to the control groups, the MDA contents of the 24 and 96 mg/L groups were significantly increased. The MDA content reached a peak at 72 h in the 24 mg/L group and 48 h in the 96 mg/L group and then decreased. 

The response of T-SOD activity (Figure 1B) showed no significant differences at any point in time in the control groups (*p* > 0.05). In comparison, the SOD activity of the 6 and 24 mg/L groups first showed a significant increase and then decreased, with peaks at 48 and 12 h, respectively. In the 96 mg/L group, SOD activity first decreased from 0 to 48 h and then increased from 48 to 96 h.

The response of the CAT activity is shown in Figure 1C. There were no significant differences in the CAT activity between each time point in the control and 6 mg/L groups. Compared to the control groups, the CAT activity of the 24 and 96 mg/L groups showed a significant increase, the CAT activity first increased and then decreased, and the activity reached a peak at 48 and 72 h, respectively.

The response of the GSH content is shown in Figure 1D. The GSH content of all treatment groups was significantly higher than that of the control group. There were no significant differences in the GSH content between each time point in the control and 6 mg/L groups. Compared to the control groups, the CAT activity of the 24 and 96 mg/L groups was significantly higher, the CAT activity increased first and then decreased, and the activity reached a peak at 48 and 72 h, respectively.

### 2.2. Pe-Cu/Zn SOD Sequence Analysis and Protein Structure

The obtained total length of *Pe-Cu/Zn SOD* cDNA was 857 bp, including 75 bp 5 ‘UTR, 323 bp 3’ UTR, and 459 bp open reading frame. There was a tailed signal sequence AATAAA upstream of the Polya tail (Figure 2). The open reading frame encoded 152 amino acids. The predicted molecular weight of the *Pe*-*Cu/Zn SOD* protein was approximately 15.6 KD and the theoretical isoelectric point was 5.65.

### 2.3. Sequence Alignment and Phylogenetic Analysis

The predicted aa sequences of the *Pe-Cu/Zn SOD* proteins were compared and aligned with their homologs in other species. The results showed that the Cu/Zn SOD sequence was highly conserved, and the similarity of *Pe*-*Cu/Zn SOD* with *Crassostrea gigas* and *Schistosoma japonicum* was 78.8% and 72.5%, respectively. In addition, *Pe*-*Cu/Zn SOD* conserved the family characteristic sequences GFHVHEFGDNT and GNAGGRLACGVI (Figure 3).

Additionally, we analyzed the phylogenetic relationships of Cu/Zn SOD using a neighbor-joining tree. The results showed that *Pe*-*Cu/Zn SOD* was located in the invertebrate evolutionary branch of icCu/Zn SOD and was far away from the evolution of ecCu/Zn SOD and icCu/Zn SOD in vertebrates (Figure 4).

### 2.4. Structural Characteristics of the Pe-Cu/Zn SOD Protein

The predicted *Pe*-*Cu/Zn SOD* protein had conserved Cu^2+^- and Zn^2+^-binding sites, in which Cu^2+^ coordinated with His-45, -47, -62, and -119 while Zn^2+^ coordinated with His-62, -70, -79, and Asp 82. A pair of intrachain disulfide bonds stabilizing the enzyme structure was formed between cysteine Cys 56 and Cys 145–SH (Figure 5).

### 2.5. Tissue Expression Patterns of Pe-Cu/Zn SOD mRNA

The qPCR results showed that *Pe-Cu/Zn SOD* mRNA was widely expressed in all tissues examined, with the highest levels in the coelomic fluid tissue (Figure 6). 

### 2.6. Expression Characteristics of Pe-Cu/Zn SOD under Cd Stress

The expression levels of *Pe*-*Cu/Zn SOD* mRNA were all increased after exposure to Cd (6, 24, and 96 mg/L) compared with the control group. In all treatment groups, the expression levels of *Pe*-*Cu/Zn SOD* mRNA showed a consistent trend, which increased from 0 to 24 h. In the 6 and 96 mg/L groups, the expression levels of *Pe*-*Cu/Zn SOD* mRNA decreased from 24 to 96 h. The expression levels fluctuated in the 24 mg/L group, increasing from 48 to 72 h and then decreasing from 72 to 96 h (Figure 7). 

The response of the SOD activity is shown in Figure 8. There were no significant differences in the SOD activity at any time in the control groups (*p* > 0.05), whereas the SOD activity of the 6 and 24 mg/L groups was significantly higher at 12 and 24 h, respectively. The activity first increased and reached a peak at 24 h and then decreased to lower than that of the control group. In the 96 mg/L group, the SOD activity first increased till 48 h and then returned to normal levels. 

### 2.7. Expression and Purification of Pe-Cu/Zn SOD Protein

The recombinant plasmid of *PeCu/Zn SOD* was constructed, and the recombinant protein was induced by IPTG. The recombinant protein was expressed in the precipitate and supernatant of the broken *E. coli* cells (Figure 9). After purification, a single band with a molecular weight of approximately 21 kDa was obtained (Figure 9).

### 2.8. Cd Tolerance of Recombinant E. coli

The transformed bacteria BL21 (pET28a) and BL21 (pET28a-*Pe-Cu/Zn SOD)* were exposed to 0.2 and 1 mM Cd^2+^. The results showed that OD_600_ of the two groups was significantly different, and OD_600_ of the recombinant bacteria was significantly higher than that of the control group. In the 0.2 mM Cd treatment group, there was a significant difference in the growth status between the control group and the recombinant bacteria after IPTG induction for 2 h (*p* < 0.05), and the growth status of the recombinant bacteria was further enhanced after IPTG induction for 4 h (*p* < 0.01) while the growth advantage of the recombinant bacteria was more significant after IPTG induction for 6 and 8 h (*p* < 0.001). In the 1 mM Cd-treated group, there was a significant difference between the control group and the recombinant bacteria after 2 and 4 h of IPTG induction (*p* < 0.01). At 6 and 8 h of induction, the growth advantage of recombinant bacteria was further enhanced (*p* < 0.001), indicating that *Pe-Cu/Zn SOD* significantly improved the Cd tolerance of *E. coli* (Figure 10).

### 2.9. Regulation of Pe-Cu/Zn SOD of the ROS Content Induced by Cd in Coelomocytes

The results showed that the content of ROS in coelomocytes of *P. esculenta* was significantly increased after 96 mg/L Cd stress (*p* < 0.01), and the content of ROS in the Cd + *Pe-Cu/Zn SOD* group was significantly lower than that in the Cd group alone (*p* < 0.01), but there was no significant difference compared with that in the control group (Figure 11), indicating that *Pe-Cu/Zn SOD* had the ability to scavenge ROS, with a protective effect against Cd-induced oxidative damage. In addition, there was no significant difference in the ROS content between the Cd and Cd + PBS groups (*p* > 0.05), indicating that the injection treatment had little effect on the content of ROS in coelomocytes (Figure 11).

## 3. Discussion

### 3.1. Sequence and Protein Structure of Pe-Cu/Zn SOD

In this study, we cloned the full-length Cu/Zn SOD cDNA sequence of *P. esculenta*, which contained an 857 bp nucleotide and encoded 152 amino acids. The unstable signal ATTTA appeared in the 3′ untranslated region, which was also found in *Meretrix meretrix* [15], *Argopecten irradians*, and other marine invertebrates [16]. It has been reported that the unstable signal ATTTA may play an important role in the degradation of excess mRNA. Cu/Zn SOD can be divided into two types: exCu/Zn SOD with a signal peptide at the N-terminal and icCu/Zn SOD without a signal peptide [10]. ExCu/Zn SOD is generally located in the extracellular matrix, with a length of 176–251 amino acids; icCu/Zn SOD mainly exists in the cytoplasm and nucleus, with a length of 147–167 amino acids [17]. In this study, *Pe-Cu/Zn SOD* contained 152 amino acids without a signal peptide at the N-terminal. Phylogenetic tree analysis showed that *Pe-Cu/Zn SOD* was located in the intracellular branch of Cu/Zn SOD. 

The amino acid sequence of *Pe-Cu/Zn SOD* was compared with that of homologous proteins of other species. Indeed, Kim et al. [18] also found the conserved tag sequences GNAGGRAACGVI and GFHIHQFGDNT of the Cu/Zn SOD family in *Pe-Cu/Zn SOD*. As predicted, the residue Cu^2+^-binding sites (His-45, -47, -62, and -119) and Zn^2+^-binding sites (His-62, -70, -79, and Asp-82) were also found in *Pe*-Cu/Zn SOD. Cys-56 and Cys-145 form disulfide bonds in the *Pe-Cu/Zn SOD* protein, which plays an important role in maintaining the structural stability of the enzyme [19]. These conserved amino acids are essential for the structure and function of Cu/Zn SOD and may be involved in the stabilization of the SOD conformation under adverse environmental conditions.

### 3.2. Physiological and Biochemical Changes of P. esculenta under Cd Stress

The main toxic effect of Cd on organisms is oxidative damage [20]. As an inducer of peroxide, Cd can stimulate the production of excessive free radicals, which eventually leads to oxidative damage [21]. MDA is a product of lipid peroxidation induced by oxygen free radicals and is a recognized biomarker in marine invertebrates [22]; for example, the change in the MDA content in the serum of *Palaemon carincauda* reflects the degree of heavy metal stress [23]. It has been reported that MDA is sensitive to Cd exposure; it typically increases slowly in organisms at low Cd concentrations while under high Cd concentrations, it increases sharply for a short time and then decreases [24]. We found the same expression pattern in this study. The MDA content increased slowly in the low concentration group but increased sharply in the high concentration group, indicating that the high concentration of Cd ions is toxic to *P. esculenta*. We suggest that MDA is related to antioxidation.

SOD is also an important antioxidant enzyme, and its activity often reflects the degree of oxidative stress. Sun et al. [25] found that a low dose of Cd induced an increase in SOD enzyme activity in the visceral mass and gills of *Tegillarca granosa* Linnaeus while a high dose of Cd inhibited it. Wang [26] found that under Cd stress, the SOD activity of *Mizuhopecten yessoensis* increased and the enzyme activity decreased. Under different concentrations of Cd stress, the SOD activity in the gills of *Sinanodonta woodiana* demonstrated the rule of “low concentration induction, high concentration inhibition” [27]. Studies have shown that Cd can change the original molecular conformation of SOD by occupying Zn or Mn structural sites in the SOD protein, thus inhibiting enzyme activity [28,29]. In summary, in the early stages of Cd exposure, SOD activity increased to eliminate excessive ROS. Under high concentrations of Cd stress, a large amount of ROS accumulated in the cells and the SOD activity decreased due to the inhibition by toxic substances. Our results are consistent with these findings. The change in SOD in the coelomic fluid of *P. esculenta* showed the same pattern of hormesis at different concentrations of Cd. At low (6 mg/L) and medium (24 mg/L) Cd stress, SOD enzyme activity first increased and then decreased, indicating that a lower concentration of Cd causes the body to produce oxygen free radicals, and the body regulates this balance by producing the SOD enzyme. In the high concentration group (96 mg/L), the activity of SOD was inhibited, which indicated that when the concentration of Cd exceeded a certain threshold, the activity of SOD decreased, resulting in tissue damage. 

CAT is known to disproportionate H_2_O_2_ into water and oxygen molecules. Livingstone et al. [30] found that CAT functions in the antioxidative system of marine invertebrates. The CAT activity in hepatopancreas of *Haliotis discus hannai* significantly increased after Cd stress, indicating its role in resisting ROS [31]. After Cd exposure, CAT activity in the tissues of *Mactra veneriformis* and *Ruditapes philippinarum* was significantly increased [32,33]. In this study, CAT activity in the coelomic fluid of *P. esculenta* did not change significantly in the control group or the 6 mg/L experimental group but increased significantly in the 24 and 9–6 mg/L Cd groups. It can be inferred that a large amount of H_2_O_2_ was produced in the coelomic fluid of *P. esculenta* under high Cd stress, which induced an increase in CAT activity.

GSH is a non-enzymatic antioxidant that scavenges ROS [34,35]. The change in the GSH content reflects the redox state of the cells [21]. An increase in the GSH content after Cd stress has been found in many marine invertebrates, for example, *Neomysis awatschensis* [36]. In this study, the GSH content in coelomic fluid increased significantly after Cd stress at 24 and 96 mg/L Cd, which indicates that GSH plays an important role in resistance to Cd-induced oxidative stress.

In conclusion, we found that Cd stress changes the physiological and biochemical indices of coelomic fluid. When exposed to Cd stress, the antioxidant enzymes and antioxidative molecules of *P. esculenta* were activated to resist oxidative damage.

### 3.3. Response of Cu/Zn SOD to Cd Stress

Cadmium can induce the formation of ROS and ultimately lead to oxidative damage [37,38]. As a member of the antioxidant system, Cu/Zn SOD is sensitive to Cd exposure. Cd can combine with the sulfhydryl group of SOD, disrupting the structure of SOD, and lead to a decrease in and inactivation of SOD [39]. When organisms are exposed to Cd stress, the activities of SOD and other antioxidant enzymes are enhanced [40,41]. Therefore, Cu/Zn SOD is a biomarker of early Cd exposure [42,43,44]. In marine invertebrates, Cu/Zn SOD has been reported to be involved in the defense against Cd stress. After exposure to Cd, the expression of Cu/Zn SOD in the digestive gland of *M. veneriformis* increased sharply, indicating that Cu/Zn SOD plays a role in maintaining cellular metabolic homeostasis and protecting clams from Cd toxicity [13].

Kim et al. [14] found that the relative expression of Cu/Zn SOD mRNA in *Euplotes crassus* increased after 0.025, 0.05, and 0.1 mg/L Cd treatment, indicating that Cu/Zn SOD may participate in the protection of cells against metal and mediated oxidative stress. For example, Zheng et al. [20] found that the Cu/Zn SOD mRNA expression and enzyme activity in *Cristaria plicata* were significantly increased after Cd stress, indicating that Cu/Zn SOD may play a role in scavenging free radicals. Xie et al. [45] reported that the Cu/Zn SOD activity of *Corbicula fluminea* was significantly increased after Cd stress and then decreased, which may be related to the elimination of free radicals and inhibition of enzyme activity.

In this study, the activity of *Pe**-Cu/Zn* SOD mRNA and enzyme was detected after Cd stress. The results showed that *Pe-Cu/Zn SOD* mRNA was significantly induced after exposure to different concentrations of Cd. The activity of the enzyme increased under low concentrations (6 and 24 mg/L) of Cd but significantly decreased at high concentrations (96 mg/L). The activity of Cu/Zn SOD mRNA and enzyme changed significantly after Cd stress, indicating that *Pe-Cu/Zn SOD* is induced in response to the oxidative stress induced by Cd.

### 3.4. Antioxidant Function of Cu/Zn SOD

Cu/Zn SOD is an enzyme that can scavenge oxygen free radicals and has strong antioxidative and immunity capacity. To analyze its functions, an in vitro experiment with purified protein was performed using *P. esculenta*. Hwang et al. [46] found that Cu/Zn SOD improved the antioxidant capacity of *Candida albicans* cells and Liu et al. [47] found that *Glyphodes pyloalis*-Cu/Zn SOD improved the tolerance of hydrogen peroxide in *E. coli*. In a similar study using *Apostichopus japonicus* [3], the recombinant *E. coli* that expressed *Aj-*Cu/Zn SOD had higher viability than the control bacteria. Perera et al. [48] detected the antioxidant activity of purified Cu/Zn SOD protein of *Hippocampus abdominalis*. The results showed that *Ha-Cu/Zn SOD* could eliminate superoxide free radicals. In our study, the Cd tolerance of *E. coli* expressing *Pe-Cu/Zn SOD* was significantly increased when compared to the control, which indicated that *Pe-Cu/Zn SOD* had a protective effect against Cd stress. These results suggest that the purified *Pe-Cu/Zn SOD* protein has an antioxidative capacity. 

In order to reveal the natural functions of Cu/Zn SOD, in vivo experiments were performed. Petkau et al. [49,50] found that intravenous injection of bovine SOD into mice can significantly repair the damage caused by X-rays in red and white blood cells. Similarly, Oda et al. [51] reported that the survival rate of mice infected with the influenza virus was significantly improved by injecting SOD protein, which indicated that SOD could improve the immunity of mice. In this study, we injected purified *Pe-Cu/Zn SOD* into the body cavities of *P. esculenta*. In contrast to the control groups, *Pe-Cu/Zn SOD* significantly reduced ROS induced by Cd in coelomocytes. Therefore, we suggest that *Pe-Cu/Zn SOD* plays an important role in the response to Cd stress and oxidative stress in *P. esculenta*.

## 4. Materials and Methods

### 4.1. Samples

*P. esculenta* was obtained from Xiangshan County (121.681777 N°, 29.48704 E°), Ningbo (Zhejiang, China). The samples were kept indoors after collection. Healthy and vital samples were selected (average weight 4.5 ± 1.0 g), kept in clean natural seawater for 24 h, and inflated continuously during temporary maintenance. All experimental procedures were approved by the Animal Care and Use Committee of the Ningbo University.

### 4.2. Chemical Exposure and Sampling

The water used in this experiment was clean natural seawater with a temperature of 22 ± 5 °C and a salinity of 28‰. The experiment was carried out in a 32 × 21 × 20 cm plastic water tank, with 12 L of continuously aerated water in each tank. According to our previous experiment, the half-lethal concentration of Cd for 96 h was 192 mg/L [52]. In this experiment, according to the 96 h LC_50_ concentration gradient (0, 1/32 of 96 h LC_50_, 1/8 of 96 h LC_50_, ½ h of LC_50_), a total of 126 *P. esculenta* were used. Six *P. esculenta* were dissected at 12, 24, 48, 72, and 96 h in each group, and the coelomic fluid was collected in 2 mL RNase-free tubes. In addition, six *P. esculenta* were collected from the control group (0 mg/L) at 0 h. Body cavity fluid was stratified and stored in liquid nitrogen and then transferred to a −80 °C refrigerator for subsequent experiments.

### 4.3. Total Superoxide Dismutase, Glutathione S-Transferase Activity, and GSH and MDA Content Determination in the Supernatants

The samples were homogenized in cold physiological saline and centrifuged at 12,000 rpm for 10 min (4 °C). Afterwards, the supernatants were collected and stored at −80 °C for further experiments. The total protein content was measured using BCA protein assay kits (CW Biotech, Beijing, China) according to the manufacturer’s instructions. A total superoxide dismutase (T-SOD) assay kit (hydroxylamine method), glutathione S-transferase (GST) assay kit, reduced glutathione (GSH) assay kit, and microscale malondialdehyde (MDA) assay kit (TBA method) from Nanjing Jiancheng Bioengineering Institute were used to test the supernatant according to the manufacturer’s instructions.

### 4.4. Full-Length Complimentary DNA Cloning of Cu/Zn SOD

Total RNA was extracted using TRIzol Reagent (Invitrogen, USA). Primer Premier v5.0 was used to select appropriate primers for *Pe-Cu/Zn SOD* (Appendix A) based on the transcriptome data of our previous study (GenBank accession No. OL757513). The HiFiScript first-strand cDNA synthesis kit (Cwbio, China) was used to obtain cDNA for intermediate segment sequence cloning. A 2 × Power Taq PCR MasterMix kit (BioTeke, China) was used for the PCR reaction. The PCR procedure was as follows: 94 °C, 5 min; 30 cycles (94 °C, 30 s; 58 °C, 30 s; 72 °C, 50 s); 72 °C, 10 min.

Based on the cloned Cu/Zn SOD cDNA intermediate fragment sequence, primers for 5 rapid amplification of cDNA ends (5 RACE) and 3′ RACE were designed (Appendix A). The 5 RACE reverse transcription assay was performed using the Smart RACE cDNA amplification kit (CloneTech, USA) while the 3′ RACE reverse transcription assay was performed using the 3′-Full RACE Core Set with a PrimeScript RTase kit (Takara, China). Both assays were conducted according to our previous research [52]. The products obtained were stored at −20 °C until further analysis.

### 4.5. Sequence Alignment, Structure Prediction, and Phylogenetic Analysis

The *Pe-Cu/Zn SOD* protein primary structures were predicted using online tools (http://www.bio-soft.net/sms/ (accessed on 3 July 2020)). The molecular weights of *Pe-Cu/Zn SOD* proteins were predicted using the ExPASy ProtParam tool (http://web.expasy.org/protparam/ (accessed on 3 July 2020)). The protein sequence was aligned using Vector NT110 (Invitrogen, CA, USA). Secondary and 3-D structures were generated and analyzed using ProtParam (http://web.expasy.org/protparam/ (accessed on 3 July 2020)) and I-TASSER (http://zhanglab.ccmb.med.umich.edu/I-TASSER (accessed on 4 July 2020)). Cu/Zn SOD homologues in various species were used for comparison, and a phylogenetic tree was constructed using MEGA v5.0. The GenBank accession numbers of the Cu/Zn SOD proteins are shown in Appendix A.

### 4.6. mRNA Expression and Enzyme Activity of Pe-Cu/Zn SOD

The expression profile of *Pe-Cu/Zn SOD* was monitored using qPCR. Total RNA was extracted and reverse-transcribed into cDNA using the PrimeScript RT reagent kit (Takara, Japan). The primers used for qPCR are listed in Appendix A, and *GAPDH* primers were used as a positive control. qPCR amplification was performed using SYBR Premix Ex Taq II (Takara, Japan). qPCR was conducted at 95 °C for 4 min, followed by 40 amplification cycles (10 s at 95 °C, 15 s at 60 °C, and 15 s at 72 °C). The comparative ΔΔCt method was used to analyze the relative expression levels of *Pe-Cu/Zn SOD*. The relative mRNA expression levels are presented as mean ± standard deviation (*n* = 6). The data were analyzed using one-way analysis of variance with SPSS v20.0, and statistical significance was defined as *p* < 0.05. The supernatants mentioned in Section 2.2 were used to determine SOD activity. *Pe-Cu/Zn SOD* activity was detected using an SOD assay kit purchased from Jiancheng Bioengineering (Nanjing, China) and calculated using the formula based on the absorbance values. SOD activity was expressed as U/mg protein.

### 4.7. Recombinant Protein Expression and Purification

Based on the cloned *Pe-Cu/Zn SOD* cDNA ORF sequence, the forward primer *Cu/Zn SOD*-F with a BamH I restriction site and the reverse primer *Cu/Zn SOD*-R with a Xho I restriction site were used to amplify the coding region of *Pe-Cu/Zn SOD* (Appendix A). The *Pe-Cu/Zn SOD* ORF was amplified using 2 × Super Pfx MasterMix (CWBIO). The *PeCu/Zn SOD* ORF and pET-28a (+) plasmids (Novagen) were double-digested and then ligated using T4 DNA ligase (TaKaRa) to obtain the pET28a-MT recombinant plasmid, which was then transferred to Trans5a Chemically Competent Cell (TransGen Biotech) and sequenced. The pET28a-MT recombinant plasmid was extracted using the Plasmid Extraction Mini Kit (Solarbio) and then transferred to Transetta (DE3) Chemically Competent Cell (TransGen Biotech) to obtain pET28a-*PeCu/Zn SOD* -DE3 recombinant *E. coli*.

pET28a-*PeCu/Zn SOD*-DE *E. coli* was expanded in liquid LB medium (+kanamycin) at 37 °C and 200 rpm until OD600 reached 0.4–0.6. Isopropyl-β-d-1-thiogalactopyranoside (IPTG) was added to the final concentration of 1 mmol/L, and then incubated for 8 h to induce protein expression. The recombinant protein was found to be expressed mainly in inclusion bodies by sodium dodecyl sulfate polyacrylamide gel electrophoresis (SDS-PAGE) analysis and purified using a His-tagged protein purification kit (Inclusion Body Protein, CWBIO). Using dialysis membranes MD44 (3500D, Solarbio), the purified proteins were sequentially dialyzed once in 50 mM PBS containing 6, 4, 2, and 1 M urea and, finally, twice in 50 mM PBS without urea for 12 h each.

### 4.8. Cd Tolerance of Recombinant E. coli

The control (pET28a) and recombinant BL21-expressing bacteria (pET28a-*Pe-Cu/Zn SOD*) were cultured at 37 °C and 200 rpm until OD_600_ reached approximately 0.4. IPTG at a final concentration of 1 mM was added to induce the expression of *Pe-Cu/Zn SOD*. At the same time, 0.2 and 1 mM CdCl_2_ were added. The shaking table culture was continued for 8 h and the OD_600_ value was determined every 2 h for each of the 3 parallel experiments conducted.

### 4.9. Analysis of the Pe-Cu/Zn SOD Protection Function In Vivo

Three treatment groups (Cd, Cd + *Pe*Cu/Zn SOD, and Cd + PBS) were set up. Six *P. esculenta* samples were treated with 96 mg/L Cd^2+^ for 24 h in each group. In the Cd + *Pe-Cu/Zn SOD* group, 100 µL of soluble *Pe-Cu/Zn SOD* purified protein was injected into the body cavity of *P. esculenta* before Cd treatment, whereas in the Cd + PBS group, it was injected with 100 µL of 1 × PBS. In the control group, *P. esculenta* was placed in clean natural seawater without Cd treatment.

### 4.10. Data Analysis

SPSS 20.0 (IBM company, Armonk, NYC, USA)and Excel software (Microsoft company, Redmond, WA, USA) were used for statistical analysis. The experimental groups were compared using one-way ANOVA and Duncan’s tests. The differences between the groups were analyzed and plotted using Graphpad software 7.0 (Graphpad software company, San Diego, CA, USA).

## 5. Conclusions

We report that the toxic effects of different concentrations of Cd (6, 24, 96 mg/L) on *P. esculenta* caused significant changes in the antioxidative indexes, T-SOD, GST, GSH, and MDA. These results indicated that Cd induced oxidative stress in *P. esculenta*. We cloned the full-length *Pe-Cu/Zn SOD* and identified it as an icCu/Zn SOD. The qPCR results showed that *Pe-Cu/Zn SOD* mRNA was expressed widely at the highest levels in the coelomic fluid. This significant increase after Cd exposure indicated that *Pe-Cu/Zn SOD* featured in the stress response. We obtained *Pe-Cu/Zn SOD* recombinant protein and found that it enhanced the heavy metal tolerance of *E. coli*. In vivo assays confirmed that *Pe-Cu/Zn SOD* recombinant protein exhibited an antioxidative activity and free radical scavenging ability, suggesting that *Pe-Cu/Zn SOD* could chelate heavy metal ions and scavenge reactive oxygen free radicals. We suggest that *P. esculenta* can be used as a bioindicator to evaluate heavy metal pollution.

## Figures and Tables

**Figure 1 ijms-23-12136-f001:**
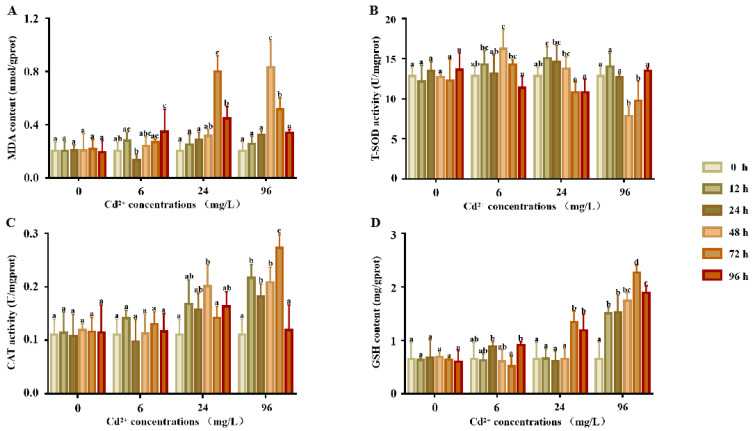
T-SOD, GST activity, and GSH and MDA contents in the supernatants of *P. esculenta* after Cd exposure. (**A**) The content of MDA in supernatants after Cd exposure. (**B**) The activity of total superoxide dismutase (T-SOD) in the supernatants after Cd exposure. (**C**) The activity of CAT in the supernatants after Cd exposure. (**D**) The content of GSH in the supernatants after Cd exposure. The color of the columns shows the different experimental times and the abscissa shows the Cd^2+^ concentrations. Lowercase letters indicate the significant differences (*p* < 0.05) of the different concentration groups at the same time (mean ± sd, *n* = 6).

**Figure 2 ijms-23-12136-f002:**
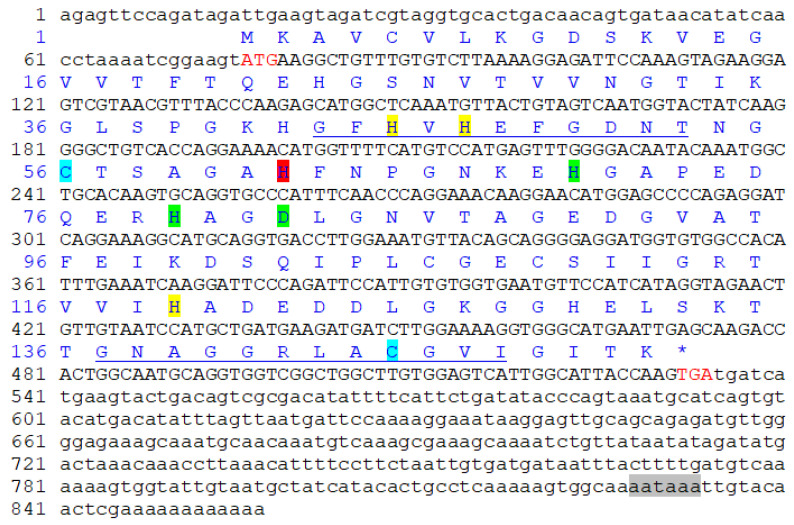
*Pe-Cu/Zn SOD* full-length cDNA and amino acid sequence. The red letters ATG and TGA indicate the start and stop codons, respectively. The highlights represent different binding sites: yellow = Cu^2+^-binding site; green = Zn^2+^ binding site; red = Cu^2+^- and Zn^2+^-binding site; blue = cysteine site; the underline represents the Cu/Zn SOD family tag sequence; and the gray shade represents the 3′ terminal tailing signal. * represents the termination codon.

**Figure 3 ijms-23-12136-f003:**
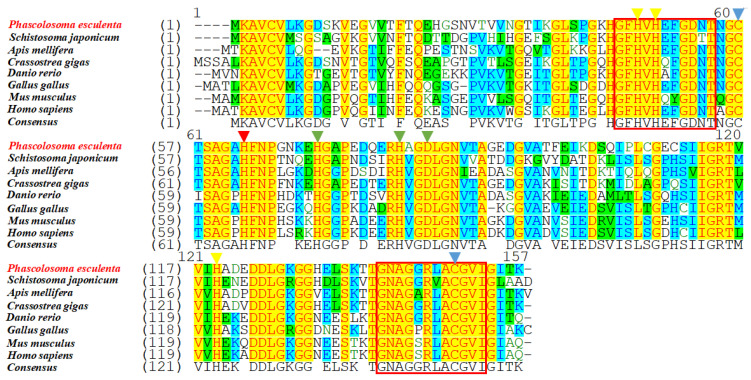
Multiple sequence alignment of Cu/Zn SOD homologous proteins. The yellow triangles indicate the Cu^2+^-binding sites, the green triangles indicate the Zn^2+^-binding sites, the red triangle indicates the Cu^2+^- and Zn^2+^-binding sites, the blue triangles indicate the cysteine sites, and the red box indicates the Cu/Zn SOD family tag sequence. The consensus and identity positions of the *Pe-Cu/Zn SOD* sequence with *H. sapiens*, *M. musculus*, *G. gallus*, *D. rerio*, *C. gigas*, *A. mellifera*, and *S. japonicum* were 68.8% and 63.0%, 68.8% and 63.0%, 71.8% and 66%, 68.8% and 65.6%, 78.8% and 76.3%, 70.8% and 63.6%, and 72.5% and 64.1%, respectively.

**Figure 4 ijms-23-12136-f004:**
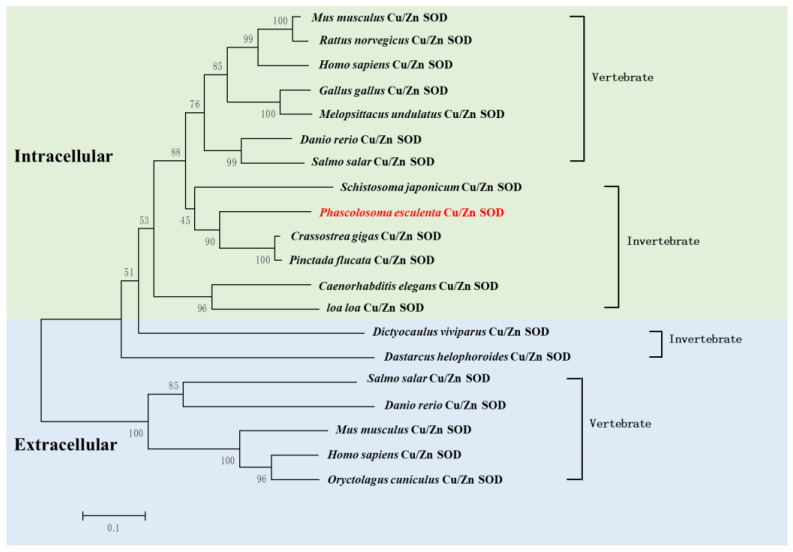
Phylogenic analysis of *Pe*-Cu/Zn SOD. The evolutionary tree was constructed using MEGA5.1 software. *P. esculenta* is shown in red font. *Pe-Cu/Zn SOD* is located in the invertebrate clade of intracellular Cu/Zn SOD. Scale bar: 0.1 of the branch length value.

**Figure 5 ijms-23-12136-f005:**
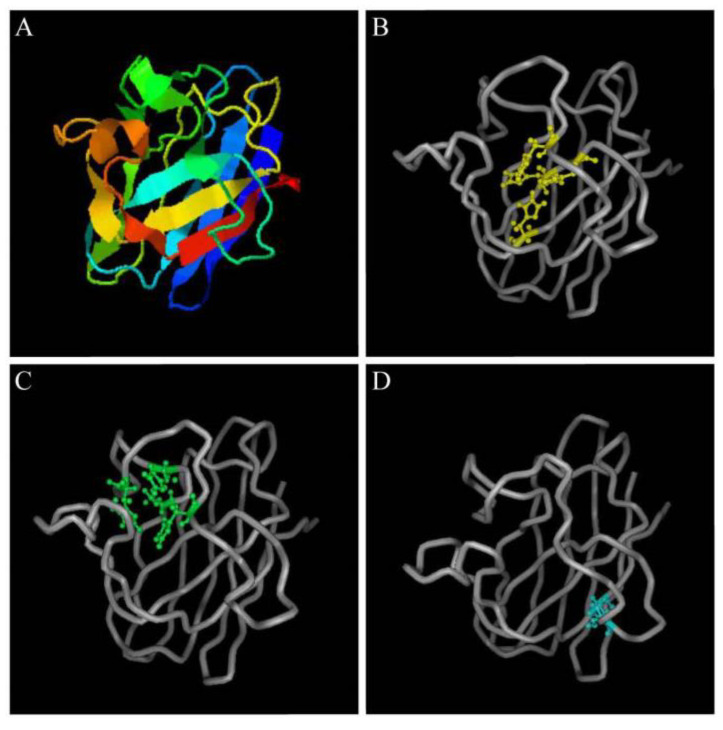
Structural prediction of *Pe-Cu/Zn SOD* proteins. (**A**) Predicted tertiary structure of *Pe-*Cu/Zn SOD; (**B**) Cu^2+^-binding sites; (**C**) Zn^2+^-binding sites; and (**D**) two cysteine sites.

**Figure 6 ijms-23-12136-f006:**
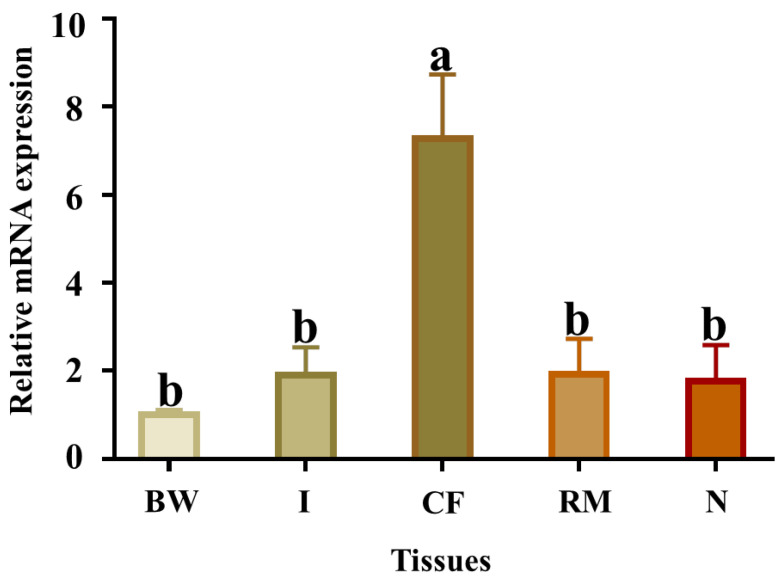
Relative abundance of *Pe-Cu/Zn SOD* mRNA detected by qPCR in different tissues. GAPDH was used as a positive control. BW, body wall; CF, coelom fluid; I, intestine; N, nephridium; RM, retractor muscle. Lowercase letters indicate a significant difference (*p* < 0.05) between different tissues (mean ± standard deviation, *n* = 6).

**Figure 7 ijms-23-12136-f007:**
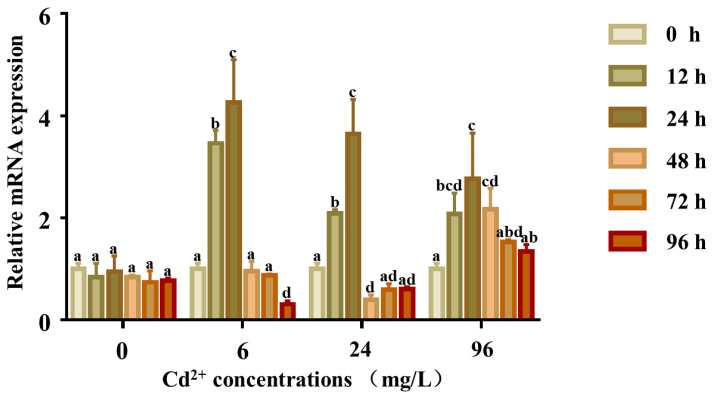
Expression of the *Pe-Cu/Zn SOD* gene in coelom fluid under Cd^2+^ stress. The expression level of *Pe-Cu/Zn SOD* mRNA in the coelom fluid after Cd exposure. GAPDH served as an internal control, and each data point represents the average fold change relative to the *Pe-Cu/Zn SOD* mRNA expression in the 0 mg/L group at 24 h, and the lowercase letters indicate significant differences (*p* < 0.05) between the data of different concentration groups at the same time (mean ± SD, *n* = 6).

**Figure 8 ijms-23-12136-f008:**
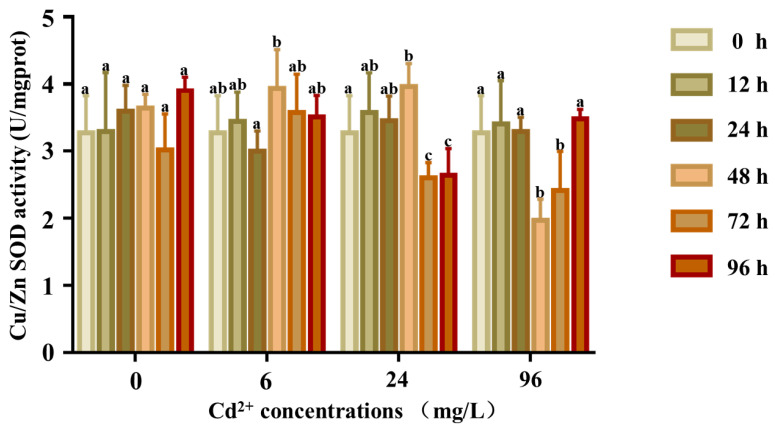
Cu/Zn SOD activity in the coelom fluid of *P. esculenta* under Cd^2+^ stress. The activity of Cu/Zn SOD in coelom fluid after Cd exposure. The color of the columns shows the different experimental times, and the abscissa shows the Cd^2+^ concentration. Lowercase letters indicate significant differences (*p* < 0.05) in different concentration groups at the same time (mean ± SD, *n* = 6).

**Figure 9 ijms-23-12136-f009:**
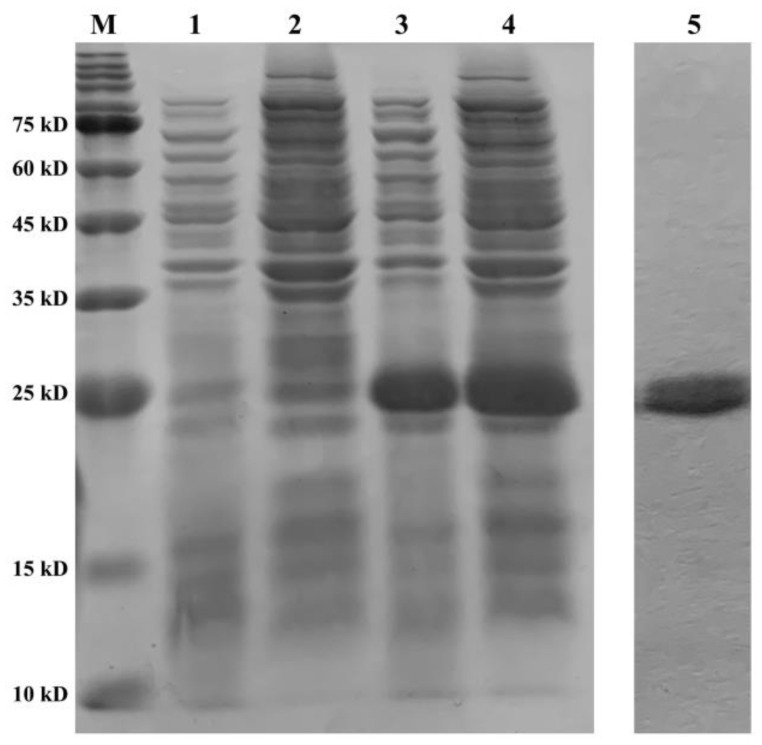
Purification and validation of the recombinant protein. Line M shows the protein marker; line 1 shows the supernatant of BL21 (pET28a) at 8 h; line 2 shows the precipitate of BL21 (pET28a) at 8 h; line 3 shows the supernatant of BL21 (pET28a-*Pe-Cu/Zn SOD)* at 8 h; line 4 shows the precipitate of BL21 (pET28a-*Pe-Cu/Zn SOD)* at 8 h; and line 5 shows the purified protein *Pe-*Cu/Zn SOD.

**Figure 10 ijms-23-12136-f010:**
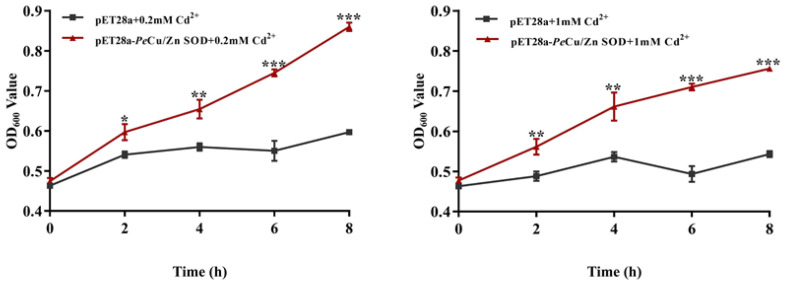
Effects of metal ions on the growth of *E. coli.* This indicates that the growth status of pET28a-*Pe-Cu/Zn SOD E. coli* was significantly higher than that of pET28a-DE3 *E. coli* (*p* < 0.5). All data is expressed as the mean ± standard deviation (*n* = 3). *: *p* < 0.05, ****: *p* < 0.01, and *****: *p* < 0.001.

**Figure 11 ijms-23-12136-f011:**
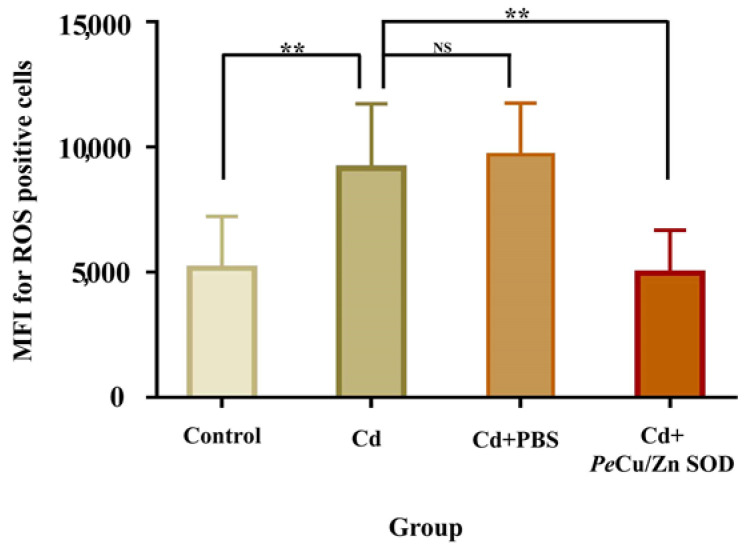
Regulation of *Pe-Cu/Zn SOD* of the ROS content induced by cadmium in coelomocytes. The ROS content in the coelomic fluid cells of *P. esculenta* was significantly increased under Cd stress; the ROS content in the Cd + *Pe-Cu/Zn SOD* group was significantly decreased compared with that in the Cd group alone and showed no significant difference with the control group. There was no significant difference between the Cd and Cd + PBS groups. All data are expressed as the mean ± standard deviation (*n* = 6). **: *p* < 0.01, NS: no significant difference (*p* > 0.05).

## Data Availability

The authors declare that all the data supporting the findings of this study are available within this article.

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
