# Peer review of "Characterization of Copper/Zinc Superoxide Dismutase Activity on Phascolosoma esculenta (Sipuncula: Phascolosomatidea) and Its Protection from Oxidative Stress Induced by Cadmium"

_ijms, 2022, doi:10.3390/ijms232012136_

Round 1

Reviewer 1 Report

Dear authors,

the manuscript deals with an important and interesting issue. However, some changes are need prior to its publication.

First of all, the species name Phascolosoma esculenta is not valid and should be corrected in the whole manuscript, and in the title, the author's name should also be added (i.e.  Phascolosoma arcuatum (Gray, 1828)), as well as the animal group (Sipunculida). Keywords should be updated and from those proposed, only the last one is ok, and the other keywords are already mentioned in the title and should be replaced by others.

In the introduction, on several places the sentence begins with Cd. If at the beggining of the sentence, the full name should be spelled out (this also refers to all other abbreviations).

Line 73- P. arcuatum is not a Sipunculus (which is another genus name), but a sipunculid.

Ln 76- response is written in italic- change to normal font

Ln 82-85- last sentences are results and do not belong to the introduction

Ln 88- add GPS coordinates of the site

Ln 89- P. arcuatum is not a plant- change this in the entire manuscript

Ln. 96- add unit to salinity, delete 3 (in cubic cm; if you write all three dimensions then it is only cm)

Ln 98. add reference to your research

Ln. 102- by strains- do you mean six animals?

Ln 103- body cavity fluid cannot be dissected

Ln 118- add reference to your research

Section 2.10. Were data tested for normality (because some data do not seem to follow normal distribution)?

In Discussion section, I suggest that the structure follows that from previous parts of thestudy, i.e. first to discuss biochemical changes and then sequence and protein structure

Ln 349, 371, 373, 384, 407- the reference is not valid

In the whole manuscript, species that are mentioned for the first time- the author and year should be added (it is currently written only for one species without a year (Ln 383).

Ln 394-395- the effects is also called "hormesis"

Ln 414- to what (which sentence) does the species Neomysis awarschensis refer to?

Species are not biomarkers- they can be bioindicators. Biomarkers are for example various enzymes (as those used in this research). Please change this accordingly.

Author Response

The manuscript deals with an important and interesting issue. However, some changes are need prior to its publication.

First of all, the species name Phascolosoma esculenta is not valid and should be corrected in the whole manuscript, and in the title, the author's name should also be added (i.e.  Phascolosoma arcuatum (Gray, 1828)), as well as the animal group (Sipunculida).

Answer: Thank you for your valuable suggestions. The Phascolosoma esculenta is an independent species which belongs to Sipuncula (class Phascolosomatidea, order Phascolosomaliformes, family Phascolomatidae). This name is universally acknowledged in study. References listed below could support this conclusion:

1.Shen W , Liu C , Ni J, et al. Effects of Low Temperature Stress on the Morphology and hsp70 and hsp90 Gene Expression of Phascolosoma esculenta[J]. Journal of Ocean University of China, 20(1):10.

2.Chen, et al. Effects of peptides from Phascolosoma esculenta on spatial learning and memory via anti-oxidative character in mice. Neuroscience Letters: An International Multidisciplinary Journal Devoted to the Rapid Publication of Basic Research in the Brain Sciences, 631(2016):30-35.

Thank you again for your kindly advices, the necessary informations were added in the tittle.

Keywords should be updated and from those proposed, only the last one is ok, and the other keywords are already mentioned in the title and should be replaced by others.

Answer: We do appreciate this valuable comment.We had re-edited the Keywords.

In the introduction, on several places the sentence begins with Cd. If at the beggining of the sentence, the full name should be spelled out (this also refers to all other abbreviations).

Answer: Thank you for your valuable suggestions. We had corrected this mistake, thank you.

Specific comments for the each section

Line 73- P. arcuatum is not a Sipunculus (which is another genus name), but a sipunculid.

Answer: We do appreciate this valuable comment.We had corrected this mistakes.

Ln 76- response is written in italic- change to normal font

Answer: We are sorry for our carelessness, we had corrected it.

Ln 82-85- last sentences are results and do not belong to the introduction

Answer: Thank you for your valuable suggestions. We had re-edited those sentences.

Ln 88- add GPS coordinates of the site

Answer: Thank you for your valuable suggestions. We added the GPS sites in manuscript now.

Ln 89- P. arcuatum is not a plant- change this in the entire manuscript

Answer: We are sorry for our carelessness, we had corrected it.

Ln. 96- add unit to salinity, delete 3 (in cubic cm; if you write all three dimensions then it is only cm)

Answer: We are sorry for our carelessness, we had corrected those mistakes.

Ln 98. add reference to your research

Answer: Thank you for your valuable suggestions. We had added the reference in manuscript.

Ln. 102- by strains- do you mean six animals?

Answer: We are sorry for our inaccurate description, we had re-edited this sentence.

Ln 103- body cavity fluid cannot be dissected

Answer: We are sorry for our inaccurate description, we changed “dissected” into “stratified”.

Ln 118- add reference to your research

Answer: Thank you for your valuable suggestions. We had added the reference in manuscript.

Section 2.10. Were data tested for normality (because some data do not seem to follow normal distribution)?

Answer: Thank you for your valuable suggestions. All the data in this study had tested for normality, we would try our best to avoid the unnormal distribution in our further work, thank you again for your kindly advice.

In Discussion section, I suggest that the structure follows that from previous parts of the study, i.e. first to discuss biochemical changes and then sequence and protein structure

Answer: Thank you for your valuable suggestions. All the sections (3.2-3.4) in Discussion are discused around the Cd stress except section 3.1, so we put this section 3.1 in front of other sections, thank you for your kindly advices.

Ln 349, 371, 373, 384, 407- the reference is not valid

Answer: We are sorry for our carelessness, we had corrected those mistakes in the paper.

In the whole manuscript, species that are mentioned for the first time- the author and year should be added (it is currently written only for one species without a year (Ln 383).

Answer: Thank you for your valuable suggestions, we found almost all the papers in this journal tend to omit the authors and years of species, we are truly agreed with your valuable suggestions, we would re-edited this after discussing with the editor.

Ln 394-395- the effects is also called "hormesis"

Answer: Thank you for your valuable suggestions, we had corrected it.

Ln 414- to what (which sentence) does the species Neomysis awarschensis refer to?

Answer: We are sorry for our carelessness, it refer to the setence “An increase in GSH content after Cd stress has been found in many marine invertebrates”. We had corrected this mistakes in manuscript.

Species are not biomarkers- they can be bioindicators. Biomarkers are for example various enzymes (as those used in this research). Please change this accordingly.

Answer: Thank you for your valuable suggestions, we had corrected it.

Reviewer 2 Report

The manuscript reports a series of well-organized, well-conducted and well-described trials to support the statement that Phascolosoma esculenta can act as a bioindicator to detect Cd pollution in marine environments; the curiosity that remains unsatisfied is "why just the Phascolosoma esculenta?" what were the reasons for getting to choose this organism as a candidate for the biomonitoring plans of heavy metal pollution? For the rest, the manuscript presents a few minor observations which the authors will be able to quickly remedy. Bibliography is accurate and updated, even if “Error! Reference source not found” in the main text are sentences that require amendment.

30 and 483: maybe the authors intended “bioindicator” …

44: please, check [0] in references

63-64: please, standardize definition, explaining icCu/Zn SOD as intracellular as ecCu/Zn SOD is indicated as extracellular

73: Please, explain the economic importance of sipuncula, used as a food in China and southeast Asia but unknown in the rest of the world

88-90: please, explain unclear sentences: P. esculenta are objects (“samples” and not “animals”), animals or vegetables (“healthy and vital plants”)?

95: “clean sea water” was tested for Cd and/or other pollutants prior to be used for the trial? Why didn’t the authors choose a synthetic controlled marine water? 

224: bars in graphs of fig.1 are difficult to distinguish, unless you do not zoom at 200%: is it possible to change them in a clearer graphic?

349, 371-2, 373-4, 384, 407: will the authors amend these sentences?

379-380: “the antioxidation of antioxidant enzymes” is uncomfortable to read, please rephrase

413-4: “An increase in GSH content after Cd stress has been found in many marine invertebrates. Neomysis awatschensis [37]”: please, check the sentence

456: “purified Hippocampus abdominalis” is an unclear statement: please, clarify the purified extracts of the seahorse

Author Response

The manuscript reports a series of well-organized, well-conducted and well-described trials to support the statement that Phascolosoma esculenta can act as a bioindicator to detect Cd pollution in marine environments;

the curiosity that remains unsatisfied is "why just the Phascolosoma esculenta?" what were the reasons for getting to choose this organism as a candidate for the biomonitoring plans of heavy metal pollution?

Answer: Thank you for your valuable suggestions. P. esculenta live in intertidal mudflats, the stably life cycle made this species countable. In addtion, they are sensitive to the changes of environmental conditions, which could reflect the pollution status of their habitats. Thus, P. esculenta are the potential bioindicator to monitor the heavy metal pollution in marine mudflats. For this reasons, we choose this organism as a candidate for the biomonitoring plans of heavy metal pollution. We had re-edited the text for easy reading.

For the rest, the manuscript presents a few minor observations which the authors will be able to quickly remedy. Bibliography is accurate and updated, even if “Error! Reference source not found” in the main text are sentences that require amendment.

Answer: We do appreciate this valuable comment.We had checked the whole manuscript and corrected the mistakes, we are sorry for our carelessness.

Specific comments for the each section

30 and 483: maybe the authors intended “bioindicator” …

Answer: Thank you for your valuable suggestions, we had corrected this mistakes.

44: please, check [0] in references

Answer:We are sorry for our carelessness, we had corrected this mistake.

63-64: please, standardize definition, explaining icCu/Zn SOD as intracellular as ecCu/Zn SOD is indicated as extracellular

Answer: We do appreciate this valuable comment. We added the necessary information in the manuscript.

73: Please, explain the economic importance of sipuncula, used as a food in China and southeast Asia but unknown in the rest of the world

Answer: Thank you for your valuable suggestions. P. esculenta is an important resource for fisheries because they are highly edible and have highly pharmaceutical values. In this paper, we focus on its role as a bioindicator, so we deleted the “economic” in the text with careful consideration.

88-90: please, explain unclear sentences: P. esculenta are objects (“samples” and not “animals”), animals or vegetables (“healthy and vital plants”)?

Answer: We are sorry for our carelessness, P. esculenta are animals, we had corrected it in the manuscript.

95: “clean sea water” was tested for Cd and/or other pollutants prior to be used for the trial? Why didn’t the authors choose a synthetic controlled marine water? 

Answer: Thank you for your valuable suggestions. The “clean sea water” was obtained from the habitats of P. esculenta. In order to mitigate the psychological stress reactions, instead of using synthetic controlled marine water, we choosed“clean sea water”.

224: bars in graphs of fig.1 are difficult to distinguish, unless you do not zoom at 200%: is it possible to change them in a clearer graphic?

Answer: Thank you for your valuable suggestions. We are truely sorry for this.We think too many groups in Fig 1 caused the bars in graphs of fig.1difficult to distinguish. We would discussed it with the editor, maybe we could divide this fig into two pictures.

349, 371-2, 373-4, 384, 407: will the authors amend these sentences?

Answer: We are sorry for our carelessness, we had corrected those mistakes

379-380: “the antioxidation of antioxidant enzymes” is uncomfortable to read, please rephrase

Answer: Thank you for your valuable suggestions. We had edited this sentence.

413-4: “An increase in GSH content after Cd stress has been found in many marine invertebrates. Neomysis awatschensis [37]”: please, check the sentence

Answer: We are sorry for our carelessness.We had edited this sentence.

456: “purified Hippocampus abdominalis” is an unclear statement: please, clarify the purified extracts of the seahorse

Answer: Thank you for your valuable suggestions. We had edited this sentence.
